# Food Addiction in Eating Disorders and Obesity: Analysis of Clusters and Implications for Treatment

**DOI:** 10.3390/nu11112633

**Published:** 2019-11-03

**Authors:** Susana Jiménez-Murcia, Zaida Agüera, Georgios Paslakis, Lucero Munguia, Roser Granero, Jéssica Sánchez-González, Isabel Sánchez, Nadine Riesco, Ashley N Gearhardt, Carlos Dieguez, Gilda Fazia, Cristina Segura-García, Isabel Baenas, José M Menchón, Fernando Fernández-Aranda

**Affiliations:** 1Department of Psychiatry, University Hospital of Bellvitge-IDIBELL, L’Hospitalet de Llobregat, 08907 Barcelona, Spain; 2CIBER Fisiopatología Obesidad y Nutrición (CIBERobn), Instituto de Salud Carlos III, L’Hospitalet de Llobregat, 08907 Barcelona, Spain; 3Clinical Sciences Department, School of Medicine, University of Barcelona, 08907 L’Hospitalet de Llobregat, Spain; 4Department of Public Health, Mental Health and Maternal-Child Nursing, School of Nursing, University of Barcelona, 08907 L’Hospitalet de Llobregat, Spain; 5Toronto General Hospital, University Health Network, Toronto, ON M5G 2C4, Canada; 6Department of Psychiatry, University of Toronto, Toronto, ON M5T 1R8, Canada; 7Department of Psychobiology and Methodology, Autonomous University of Barcelona, 08193 Barcelona, Spain; 8Department of Psychology, University of Michigan, Ann Arbor, MI 48109, USA; 9Department Physiology, Center for Research in Molecular Medicine and Chronic Diseases (CIMUS), Health Research Institute of Santiago USC-IDIS, University Santiago de Compostela, 15705 Santiago de Compostela, Spain; 10Department of Health Sciences, University “Magna Graecia” of Catanzaro, Viale Europa, 88100 Catanzaro, Italy; 11Center for Clinical Research and Treatment of Eating Disorders, Azienda Ospedaliera Universitaria Mater Domini, 88100 Catanzaro, Italy; 12Department of Medical and Surgical Sciences, University “Magna Graecia” of Catanzaro, 88100 Catanzaro, Italy; 13CIBER de Salud Mental (CIBERSAM), Instituto de Salud Carlos III, L’Hospitalet de Llobregat, 08907 Barcelona, Spain

**Keywords:** food addiction, eating disorders, bulimia nervosa, binge eating disorders, obesity, other specified feeding or eating disorders, cluster analysis

## Abstract

Food addiction (FA) has been associated with greater psychopathology in individuals with eating disorders (ED) and obesity (OBE). The current study aims to provide a better phenotypic characterization of the FA construct by conducting a clustering analysis of FA in both conditions (ED and OBE). The total sample was comprised of 234 participants that scored positive on the Yale Food Addiction Scale 2.0. (YFAS-2) (119 bulimia nervosa (BN), 50 binge eating disorder (BED), 49 other specified feeding or eating disorder (OSFED) and 16 OBE). All participants completed a comprehensive battery of questionnaires. Three clusters of FA participants were identified. Cluster 1 (dysfunctional) was characterized by the highest prevalence of OSFED and BN, the highest ED severity and psychopathology, and more dysfunctional personality traits. Cluster 2 (moderate) showed a high prevalence of BN and BED and moderate levels of ED psychopathology. Finally, cluster 3 (adaptive) was characterized by a high prevalence of OBE and BED, low levels of ED psychopathology, and more functional personality traits. In conclusion, this study identified three distinct clusters of ED-OBE patients with FA and provides some insight into a better phenotypic characterization of the FA construct when considering psychopathology, personality and ED pathology. Future studies should address whether these three food addiction categories are indicative of therapy outcome.

## 1. Introduction

Food addiction (FA) is a concept that has been of increasing scientific interest and debate. An immense body of literature within the field of eating disorder (ED) research has emerged, with whole special issues of scientific journals being dedicated to its characterization [1,2]. In addition to obvious phenomenological similarities between addiction and ED (e.g., loss of control, continued use despite negative consequences, cravings), a great number of neurobiological findings have emerged to additionally support the new concept, not only in preclinical studies, but also in humans [3,4,5,6].

Still, not all controversies have yet been resolved. Starting off as a concept to explain a potential subtype of obesity (OBE) [7,8,9,10], FA has also been associated with ED, such as bulimia nervosa (BN) [11,12,13], binge eating disorder (BED) [14,15,16], and even anorexia nervosa (AN) [17]. In a systematic review of studies on FA in non-clinical and clinical cohorts, it was especially BED that was associated with the most severe FA symptoms [15]. FA also seems to be prevalent in individuals with OBE waiting for bariatric surgery [18,19] and predicts less effective weight reduction throughout dietary intervention before surgery [20]. Interestingly, surgery-induced weight loss may lead to remission of FA [21,22].

The Yale Food Addiction Scale (YFAS) is the main instrument for the assessment of FA and has been developed to assess FA based on the known criteria used for the assessment of substance dependence, but applied for high palatable foods [23,24]. Higher scores on the YFAS have been associated with higher body mass index (BMI), binge eating episodes, impulsivity, and cravings for highly palatable food [25], as well as with neural responses similar to those found in substance use disorders [26,27,28,29].

There is scarce evidence with regard to the identification of key determinants of FA based on personality traits or ED-related symptoms and most of what is known is derived from non-clinical cohorts. In a non-clinical sample, negative urgency (irrational acting in aversive affective states) and low levels of task persistence (lack of perseverance) were shown to be significantly and directly associated with FA and FA mediated their association to BMI [30]. In another study in undergraduates, negative urgency, impulsivity when under distress, and emotion dysregulation positively predicted symptom count on the YFAS [31]. Similar findings were shown in a clinical ED cohort, although negative urgency appeared to be the only independent predictor for FA, while self-directedness and emotion dysregulation predicted negative urgency and were highly related to ED-related symptomatology, but not to food addiction itself [32,33]. In individuals with OBE awaiting bariatric surgery, FA was associated with personality traits such as neuroticism, impulsivity, and alexithymia [34], but also more emotion dysregulation, more harm avoidance, and less self-directedness [35].

FA has also been associated with a positive screen for more severe variants of ED-related psychopathology [7] as well as with a positive screen for major mental health symptoms, major depressive episode, anxiety, early life adversities, such as psychological and sexual abuse, and an overall reduced quality of life [36,37,38]. Finally, female gender was a predictor of severe food addiction [36] and high reward sensitivity was significantly associated with more severe FA symptoms in females [39].

Due to these meaningful interrelationships in the literature between FA and personality traits, as well as psychopathology, it is important to consider the association of FA with these factors in an integrated way (rather than looking at each construct in isolation). Evaluating how personality and psychopathology cluster within the FA construct could lead to a better understanding of potential distinct phenotypes within FA that could have different clinical profiles.

Based on this premise, the aim of the present paper was to explore empirically the severity of clusters of FA-positive (FA+) participants based on psychopathological symptomatology (namely ED-related psychopathology and general psychopathology) and personality traits, and to investigate how the clinical features and diagnosis of ED and OBE were distributed among them. This is the first study that attempts this type of analysis in order to identify subgroups among participants with positive FA in order to provide a better phenotypic characterization of the FA construct. We hypothesized that, despite overlaps, patients with ED would predominantly fit into a cluster characterized by a more severe ED-related psychopathology that would be different than the cluster predominantly found among individuals with OBE. We then evaluated whether there were differences by ED subtypes, with a subgroup of patients (basically BED, but also BN) who are more similar to participants with OBE. We also hypothesized that, among others, ED-related severity, general psychopathology, and personality traits would be important determinants of the FA phenotypes. The rationale for performing this kind of analysis was to gain insight into the possibility of different FA clusters that would ideally translate into future symptom-targeted treatments for FA phenotypes in individuals with OBE and for those suffering from ED.

## 2. Materials and Methods

### 2.1. Participants

From an initial sample of 395, the final participants of the current study were 234 females who scored positive for the FA (based on YFAS 2.) and who also had a diagnosis of ED or OBE (119 with BN, 50 with BED, 49 with other specified feeding or eating disorder (OSFED), and 16 with OBE). All participants in the OBE group (*n* = 16) as well as *n* = 42 participants in the BED group (84.0%), *n* = 27 in the BN group (22.7%), and *n* = 2 in the OSFED group (4.1%) had a BMI higher than 30. All participants included in the study were consecutively referred for assessment and treatment at the Unit of Eating Disorders of the Department of Psychiatry of the University Hospital of Bellvitge in Barcelona, between May of 2016 and November 2018, diagnosed according to the DSM-5 [40] criteria, and were between 18 and 60 years old.

Male participants referred for the Unit in that time period were excluded from the study, as the number of them in our sample was too small for meaningful statistical comparisons (*n* = 21; four with BN, eight with BED, two with OSFED, and seven with OBE). Considering the controversial results of the presence of FA in AN [17], as well as the characteristic fears and cognitive distortions about food and weight involved in AN that may influence patients understanding of what is considered excessive food intake or abnormal eating behavior [41], patients diagnosed with AN were excluded from the study as well. Likewise, due to the different reported prevalence of FA in OBE patients [20] compared to OBE with a comorbid ED [14,15,17], the homogeneity of the sample was preserved by only inducing patients with OBE without ED (*n* = 53), and only those with positive FA were included (*n* = 16).

According to the Declaration of Helsinki, the present study was approved by the Clinical Research Ethics Committee (CEIC) of Bellvitge University Hospital (ethic approval code: PR205/17), and written informed consent was obtained from all participants. All the assessments were conducted by experienced psychologist and psychiatrists.

### 2.2. Assessment

For the assessment, anthropometric measures such as weight and height (without the participants wearing clothes or shoes) were taken to calculate the BMI (i.e., weight (kg)/height (m^2^)). In addition to clinically relevant variables (like age of onset or duration of the disorder) and sociodemographical characteristics, a battery of the Spanish-validated versions of the following instruments was used.

#### 2.2.1. Eating Disorders Inventory 2 (EDI-2)

The Eating Disorders Inventory 2 (EDI-2) [42] is a 91-item multidimensional self-report questionnaire answered on a 6-point Likert scale that assesses different cognitive and behavioral characteristics typical for ED: Drive for Thinness, Bulimia, Body Dissatisfaction, Ineffectiveness, Perfectionism, Interpersonal Distrust, Interoceptive Awareness, Maturity Fears, Asceticism, Impulse Regulation, and Social Insecurity. The validated version for the Spanish population was developed by Garner, 1998 [43], with a mean internal consistency of 0.63 (coefficient alpha).

#### 2.2.2. Symptom Checklist-90-Revised (SCL-90-R)

The Symptom Checklist-90-Revised (SCL-90-R) [44] is used to evaluate a broad range of psychological problems and symptoms of psychopathology. It consist of a 90-item questionnaire that measures nine primary symptom dimensions: Somatization, Obsession-Compulsion, Interpersonal Sensitivity, Depression, Anxiety, Hostility, Phobic Anxiety, Paranoid Ideation, and Psychoticism; and includes three global indices: global severity index (overall psychological distress), positive symptom distress index (the intensity of symptoms), and a positive symptom total (self-reported symptoms). The global severity index can be used as a summary of the test. The validation of the scale in a Spanish population [45], obtained a mean internal consistency of 0.75 (coefficient alpha).

#### 2.2.3. Yale Food Addiction Scale 2.0 (YFAS-2)

The Yale Food Addiction Scale 2.0 (YFAS-2) [24] is a 35-item self-report questionnaire for measuring FA during the previous 12 months. This original instrument (YFAS) was based on the Diagnostic and Statistical Manual of Mental Disorders (DSM-IV-TR) criteria for substance dependence and was adapted to the context of food consumption. The newer version of the instrument, YFAS-2, is based on DSM-5 Criteria and evaluates 11 symptoms. The scale produces two measurements: (a) a continuous symptom count score that reflects the number of fulfilled diagnostic criteria (ranging from 0 to 11), and (b) a food addiction threshold based on the number of symptoms (at least 2) and self-reported clinically significant impairment or distress. This final measurement allows for the binary classification of food addiction (present versus absent). Additionally, based on the revised DSM-5 taxonomy, is possible to establish severity cut-offs: mild (2–3 symptoms), moderate (4–5 symptoms), and severe (6–11 symptoms). The translation and validation of the YFAS-2 for Spanish speaking samples with ED was carried out by Granero et al. (2018) [13], showing excellent internal reliability coefficient (α = 0.94), as well as an excellent accuracy in discriminating between healthy controls and ED subsamples (κ = 0.75)

#### 2.2.4. Temperament and Character Inventory-Revised (TCI-R)

The Temperament and Character Inventory-Revised (TCI-R) [46] is a 240-item, 5-point Likert scale, questionnaire that measures four temperament dimensions (Harm Avoidance, Novelty Seeking, Reward Dependence, and Persistence) and three character dimensions (Self-Directedness, Cooperativeness, and Self-Transcendence) of personality. The Spanish validation in an adult population was carried out by Gutiérrez-Zotes et al. [47]; the internal consistency (coefficient alpha) of the scales was 0.87.

### 2.3. Statistical Analysis

Statistical analysis was carried out with SPSS23 for Windows. Empirical clusters were explored with the two-step-cluster procedure, using as indicators the ED severity level (EDI-2 scores), the global psychopathological state (SCL-90-R scores), the personality profile (TCI-R scores), and the DSM-5 diagnostic subtype. Two-step-cluster used the log-likelihood distance measure through a multinomial probability mass function for categorical variables and a normal density function for continuous variables. This clustering technique has desirable features which makes it different from traditional grouping and latent class techniques [48,49]: (a) scalability, which allows analysis of large data files by constructing a cluster-features-tree which is used as a summary of the records; (b) automatic selection of the number of clusters-classes; and (c) handling of categorical and quantitative variables. Criteria for the final model selected in this study were adequate goodness-of-fit (based on a cohesion and separation index) and adequate clinical interpretability [50]. In this study, the Silhouettes coefficient was used as a measure of the goodness of the final cluster solution. This coefficient estimates the cohesion of the elements within a cluster and the separation between the clusters, and it ranges from –1 to +1 (being the values lower than 0.30 interpreted as poor fitting, between 0.30 and 0.50 as fair fitting, and higher than 0.50 as good fitting [51]).

Comparison between clusters was based on chi-square tests (χ^2^) for categorical variables and analysis of variance (ANOVA) for quantitative criteria. Effect size was measured through Cohen’s-d coefficient for mean and proportion comparisons (low effect size was considered for |*d*| > 0.2, moderate for |*d*| > 0.5, and large for |*d*| > 0.8; [52]). The Finner’s method was used to control Type-I error due to multiple statistical comparisons (this procedure is included into the Familywise error rate stepwise procedures and offers a more powerful test than the classical Bonferroni correction) [53].

## 3. Results

### 3.1. Characteristics of the Sample

All the participants in the analyses were women who met criteria for FA positive screening score based on the YFAS-2 questionnaire *n* = 234. The distribution of the whole sample according to diagnostic group was 16 OBE (without ED) (6.8%), 50 BED (21.4%), 119 BN (50.9%), and 49 OSFED (20.9%). Most of the participants were single (62.0%, compared to 24.8% married or living with a stable partner and 13.2% separated or divorced) and achieved primary (40.6%) or secondary (44.9%) education levels. Table A1 (Appendix A) includes the distribution of the clinical profiles stratified and compared by the diagnostic subtype.

### 3.2. Cluster Composition

The best grouping structure selected for the sample of the study was the three-cluster solution, which coincided with the most optimal solution chosen by the two-step-cluster procedure. This solution obtained a Silhouettes index (0.30) into the fair range, suggesting mild-moderate evidence of the cluster structure in the data. The comparison between the largest cluster size (*n* = 90, 38.5%) and the smallest (*n* = 60, 25.6%) yielded a ratio of 1.50.

Figure 1 summarizes the results of the clustering procedure: the bar-graph with the indicator relevance for each variable (which reports how good each variable was for the grouping and can differentiate between the derived clusters: the higher the importance of the measure, the less likely it is that the variation for the variable between clusters is due to chance and the more likely it is due to underlying differences) and the centroids (which summarizes the cluster patterns for the set of variables and allows clinical interpretation of the empirical clusters). The indicator variables with the highest contribution into the clustering were SCL-90-R scales measuring the symptom levels (concretely psychotic, depressive, interpersonal sensitivity, anxiety, paranoia), followed by the EDI-2 scales measuring ED severity (impulse regulation, social insecurity, and ineffectiveness). The personality traits measured with the TCI-R obtained low relevance for the clustering, except for harm avoidance (which achieved moderate-low capacity), as well as the diagnostic subtype (which achieved poor capacity for the differentiation between the groups).

### 3.3. Comparison between the Empirical Clusters

The first part of Table 1 contains the comparison between the three empirical clusters for the civil status and the education levels. Differences between the groups only were found for the civil status: compared with the other two groups, cluster 1 had the highest proportion of single participants.

The second block of Table 1 contains the comparison between the clusters for the clinical profile. As a whole, clusters were ordered by the psychopathological state and personality profiles. Cluster 1 (*n* = 60, 25.6%) included the youngest participants (mean 28.8 years-old), with the lowest age of onset (mean 18.5 years-old), the shortest duration of the eating problems (mean 10.5 years), and the lowest BMI (mean 27.6 kg/m^2^), but with the highest severity in eating problems (the highest means in the EDI-2), the worst psychopathological state (the highest means in the SCL-90R), the highest levels in the personality traits of harm avoidance and self-transcendence, and the lowest levels in self-directedness and cooperativeness. Cluster 1 was labeled in this study as the “dysfunctional cluster”.

Cluster 2 (*n* = 90, 38.5%) included the participants with the longest duration of the eating problems (mean 14.3 years) and the highest level of food addiction (mean 9.6). The mean for BMI (30.3 kg/m2) was higher than in cluster 1, but similar to that obtained in cluster 3. The mean scores in the EDI-2 and the SCL-90R were high for participants into cluster 2, but clearly lower than values registered for cluster 1. Cluster 2 was labeled in this study as the “moderate cluster”.

Cluster 3 (*n* = 84, 35.9%) was characterized by similar mean scores in age (34.5 years-old), onset of eating problems (21.7 years) and duration of the disorder (12.6 years) than cluster 2. The most adaptive scores in the clinical profile was registered for cluster 3 (the lowest scores in the EDI-2 and the SCL-90R), as well as the highest means in the personality traits of persistence, self-directedness and cooperativeness. Cluster 3 was labeled in this study as the “functional cluster”.

Figure 2 includes the 100%-stacked bar chart with the distribution of the DSM-5 ED diagnostic subtype into each empirical cluster. OBE patients were primary included into cluster 3 (68.85), while BED patients were mostly distributed into clusters 3 and 2 (48.0% and 44.0%, respectively). Approximately half of the BN patients were into cluster 2 (52.1%), and the remaining participants into this group were distributed between cluster 1 (27.7%) and cluster 3 (20.2%). A little more than half of the OSFED patients were in cluster 3 (51%), and almost 41% in cluster 1.

As a synthesis of the results, Figure 3 contains the radar-chart comparing the empirical clusters for the main clinical variables of the study. To allow adequate interpretability, z-standardized scores were plotted.

## 4. Discussion

The aim of the present study was to explore empirical severity clusters with FA-positive (FA+) females, and to investigate whether these FA clusters differed by ED and OBE. A three-cluster structure was detected based on general psychopathology, ED severity, and personality traits. These clusters ranged from a more functional cluster, to moderate and highly dysfunctional group.

Although high FA has previously been associated with high eating psychopathology and more dysfunctional personality traits, [32], this is the first study that analyzes a potential heterogeneity within patients with FA. Although all the participants in the current study were FA+, we found that the identified clusters followed a linearity with respect to FA severity with the most dysfunctional clusters (1 and 2) having the highest FA symptoms level, and the most functional one, the lowest. For the “dysfunctional cluster”, we found a higher prevalence of OSFED and BN, both ED conditions characterized by more dysfunctional personality traits, greater impulsivity, and more general psychopathology, [54,55,56,57], as well by their worse prognosis [56]. Consistent with this literature, the cluster with more OFED and BN had the worst psychopathological state and highest severity in ED symptomatology.

Personality traits that were elevated in the most dysfunction cluster were greater difficulties in establishing goals and objectives to guide their lives (self-directedness), the highest levels of anxiety, worry, fear (harm avoidance), and being a more self-centered person (lower cooperativeness). All of these characteristics have not only been associated with FA [32,35], but also with BN, substance use disorders (SUD), and comorbidity between BN and SUD [58,59]. This cluster was also associated with higher levels of novelty seeking, which is a predictor of risky behaviors and associated with SUD and ED pathologies [59,60,61,62,63,64]. Therefore, the commonality of these extreme personality features in addictive profiles, suggests it could be important in the etiology and maintenance of FA and could be the focus of intervention efforts. Additionally, the aforementioned characteristics have been related to higher risk of dropout and poor outcome in OSFED patients with purging behaviors [56], suggesting important clinical relevance. Although the association of FA and treatment outcome was not the aim of this study, prior research suggests that higher initial FA scores have been associated with a worse prognosis in BN [65].

In sum, the severity of this dysfunctional cluster may be driven by the comorbid ED provided and characteristics associated with these disorders. Thus, in order to be successful, the treatment aimed at patients who are in this cluster should target its principal psychological and psychiatric characteristics, in addition to factors that may maintain FA. As previous literature has reported, changes in personality traits are related to an overall improvement in eating pathology [66], and interventions that could target the personality factors elevated in this cluster would likely be of benefit. Additionally, due to the severity of the psychopathological state in this cluster, pharmacological approaches to address comorbid psychiatric conditions could be also considered if it is needed.

The second cluster, the moderate cluster, was the one that presented the highest level of FA, although functioning was more adaptive than cluster 1 (i.e., lower ED severity, intermediate psychopathology levels). Therefore, we hypothesize that it is FA which mostly determines the characteristics of this cluster. Regarding the personality traits, high levels of harm avoidance and low self-directedness were present in this cluster, but at moderate levels compared to cluster 1 (the dysfunctional). Related to the ED pathology, there was a high presence of patients with BN followed by patients with BED within this cluster. Higher FA scores have been already associated with bingeing ED-subtype patients [13,14,17,67], given the tendency to consume more high-fat/high-sugar caloric food during binges episodes, which may result in a higher number of FA symptoms [17] given the similar neural responses in reward pathways modulated by dopamine by those types of food and addictive drugs [6,16,29,68,69,70]. In fact, new maintenance models of BN and BED have emerged, that take into consideration the addictive response to palatable foods [71]. Related studies have also found that similar patterns of neural underpinning of tolerance and dependence observed in SUD appear to be related to binge-type eating disorders as well [72,73]. This is consistent with the high comorbidity between SUD and binge-type ED [74,75]. Finally, another important aspect to mention is that it has been shown that individuals with BN or BED experience more frequent and more intense food cravings than persons without binge eating [76,77,78,79] and that both conditions show significantly larger food cue reactivity (self-reported craving) [80]. This intense desire or urge to eat a particular type of food also, at a neural level, resembles responses to drug cues in SUD [26]. Therefore, it could be suggested that the treatment directed to patients in this cluster should target the reward related neural processes that maintain addictive disorders. In this regard, it has been suggested that previously developed interventions for addictions could be applied to binge eating behaviors [71], such as training in the reduction of food cue reactivity in order to reduce craving [81,82] and a reduction in the intake of high-fat/high-sugar caloric food which hyper-activate reward systems.

The “adaptative” cluster presents with more functional personality traits and low levels of general psychopathology, as well as the lowest levels of FA. Thus, the FA in this cluster may be the result of different factors than patients in clusters 1 and 2, which could have important implications in the treatment. First, it is important to mention that within this cluster, there was the highest presence of patients with OBE without ED. This is consistent with prior findings that patients with OBE with a comorbid ED have a higher level of psychopathology than OBE patients without ED [83,84,85,86]. Second, this cluster presents the most functional personality traits, the lowest levels of harm avoidance and self-transcendence, and the highest in cooperativeness and self-directedness. Similar results have been found in healthy control groups when comparing them with ED and behavioral addiction patients [87,88]. Thus, BMI could be playing an important role in this cluster, considering that in our sample there are statistically significant differences between cluster 1 (the dysfunctional) and 3 (the adaptative) in BMI. It has been suggested that visceral adiposity levels could be a mediator of the relationship between middle-dorsal insula network connectivity (insula region relevant for eating behavior) and food craving [28], being that visceral fat disrupts insula coding of bodily homeostatic signals, which may boost externally driven food cravings, and also, there are positive associations between food craving and excessive overeating [89,90]. Given the characteristics of this cluster, for which no dysfunctional personality traits of severe ED symptomology or psychopathology must be addressed, it could be hypothesized that nutritional changes that would have a positive impact on a reduction of BMI could be also be beneficial for the reducing FA, through reducing craving episodes. This is consistent with previous studies that find that FA decreases significantly after bariatric surgery [91], and that the induced weight loss by the surgery resulted in the remission of FA in 93% of patients [22]. Finally, there is a moderate representation of patients with BED in this cluster, which may be due to the common co-occurrence between both conditions (OBE-BED). This may be due to the association of features such as grazing [92], craving [93,94,95], or hedonic [96] and emotional eating [97,98] with OBE and BED. Addressing these factors (and their negative consequences) could be an important focus of treatment in this FA cluster, as grazing is associated with poorer weight loss treatment outcomes in OBE [92] and craving and the use of food to regulate mood are potential triggers for overeating [41,99,100].

The results of this study should take into account the following limitations. First, the sample only included women, so the results cannot be generalized to males; for future studies, it will be important to explore if the FA cluster structure found in this study is replicated in males. Another limitation is that the size of some of the participant groups, divided by ED subtype and OBE, is small. Finally that, due to the fact that the participants were recruited in the same geographical area, the results may not be generalizable to other samples

## 5. Conclusions

The findings in the present study describe a three-cluster structure of FA-positive (FA+) participants that differ by ED and OBE profile. The clusters range from more to less functional, depending on psychopathology and personality traits. The identification of phenotypes in FA will not only increase knowledge of each cluster’s characteristics, but may allow for better individualization of treatment by identifying novel intervention targets and improving treatment outcomes. Likewise, the present study identifies future lines of research, as longitudinal studies that could analyze the predictive validity of this cluster structure on treatment outcome could be of importance.

## Figures and Tables

**Figure 1 nutrients-11-02633-f001:**
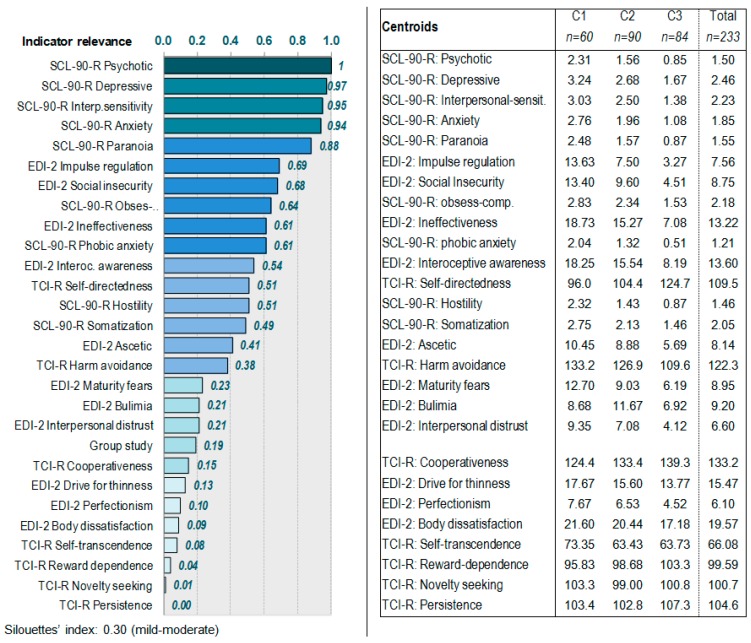
Results of the clustering procedure. Note. EDI-2: Eating disorders Inventory 2. SCL-90-R: Symptom Checklist-90-Revised. SCL-90-R Obsess-comp.: Obsession-Compulsion SCL-90-R subscale. TCI-R: Temperament and Character Inventory-Revised.

**Figure 2 nutrients-11-02633-f002:**
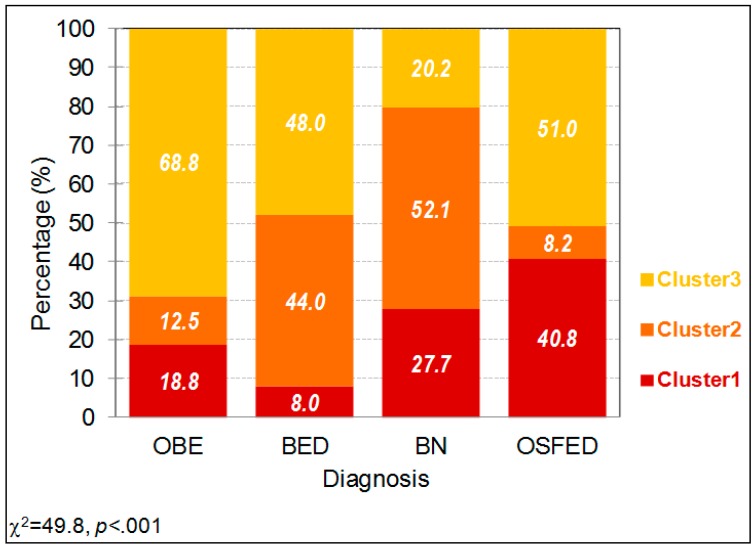
Distribution of the diagnostic subtype and the food addiction (FA) severity group within the empirical clusters. Note. OBE: obesity. BED: binge eating disorder. BN: bulimia nervosa. OSFED: other specified feeding eating disorder.

**Figure 3 nutrients-11-02633-f003:**
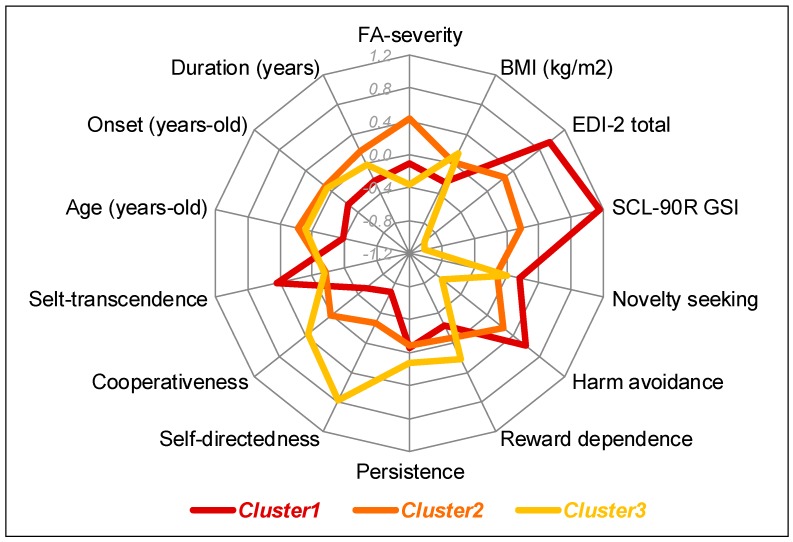
Radar-chart comparing the z-standardized mean scores between the empirical clusters Note. FA: Food addiction. BMI: Body mass index. EDI-2: Eating disorders Inventory 2. SCL-90-R GSI: Global Severity Index of the Symptom Checklist-90-Revised.

**Table 1 nutrients-11-02633-t001:** Comparison between clusters for variables of the study.

	Cluster 1		Cluster 2		Cluster 3		Pairwise Comparisons					
	*n* = 60 (25.6%)		*n* = 90 (38.5%)		*n* = 84 (35.9%)		Cl1-Cl2		Cl1-Cl3		Cl2-Cl3	
	*n*	*%*	*n*	*%*	*n*	*%*	*p*	*|d|*	*p*	*|d|*	*p*	*|d|*
Civil status—Single	46	76.7%	53	58.9%	46	54.8%	**0.049 ***	0.39	**0.026 ***	**0.51 ^†^**	0.620	0.08
Married—Couple	10	16.7%	22	24.4%	26	31.0%		0.19		0.34		0.15
Separated—divorced	4	6.7%	15	16.7%	12	14.3%		0.32		0.25		0.07
Education Primary	29	48.3%	33	36.7%	33	39.3%	0.164	0.24	0.083	0.18	0.688	0.05
Secondary	27	45.0%	43	47.8%	35	41.7%		0.06		0.07		0.12
University	4	6.7%	14	15.6%	16	19.0%		0.29		0.38		0.09
	*Mean*	*SD*	*Mean*	*SD*	*Mean*	*SD*	*p*	*|d|*	*p*	*|d|*	*p*	*|d|*
Age (years-old)	28.77	11.05	35.66	12.99	34.54	12.13	**0.001 ***	**0.57 ^†^**	**0.006 ***	**0.50 ^†^**	0.546	0.09
Age of onset (years-old)	18.50	9.17	21.97	10.71	21.66	9.10	**0.034 ***	0.35	0.057	0.35	0.837	0.03
Duration (years)	10.49	9.10	14.30	10.20	12.64	8.72	**0.016 ***	0.39	0.177	0.24	0.246	0.17
YFAS-2: total criteria	8.22	2.66	9.59	1.80	7.55	2.77	**0.001 ***	**0.61 ^†^**	0.102	0.25	**0.001 ***	**0.87 ^†^**
BMI (kg/m^2^)	27.55	11.13	30.29	9.08	31.48	10.99	0.114	0.27	**0.025 ***	0.36	0.447	0.12
EDI-2: Drive for thinness	17.67	2.84	15.60	4.82	13.77	4.83	**0.005 ***	**0.52 ^†^**	**0.001 ***	**0.98 ^†^**	**0.007 ***	0.38
EDI-2: Body dissatisfaction	21.60	5.61	20.44	6.52	17.18	7.06	0.288	0.19	**0.001 ***	**0.69 ^†^**	**0.001 ***	**0.51 ^†^**
EDI-2: Interoceptive awar.	18.25	5.77	15.54	5.33	8.19	5.12	**0.003 ***	**0.52 ^†^**	**0.001 ***	**1.84 ^†^**	**0.001 ***	**1.41 ^†^**
EDI-2: Bulimia	8.68	5.30	11.67	3.85	6.92	4.69	**0.001 ***	**0.64 ^†^**	**0.023**	0.35	**0.001 ***	**1.11 ^†^**
EDI-2: Interpers.distrust	9.35	5.23	7.08	4.76	4.12	3.89	**0.003 ***	**0.52 ^†^**	**0.001 ***	**1.14 ^†^**	**0.001 ***	**0.68 ^†^**
EDI-2: Ineffectiveness	18.73	6.57	15.27	6.06	7.08	4.50	**0.001 ***	**0.55 ^†^**	**0.001 ***	**2.07 ^†^**	**0.001 ***	**1.53 ^†^**
EDI-2: Maturity fears	12.70	5.69	9.03	5.37	6.19	5.18	**0.001 ***	**0.66 ^†^**	**0.001 ***	**1.20 ^†^**	**0.001 ***	**0.54 ^†^**
EDI-2: Perfectionism	7.67	4.63	6.53	4.21	4.52	3.65	0.101	0.26	**0.001 ***	**0.75 ^†^**	**0.002 ***	**0.51 ^†^**
EDI-2: Impulse regulation	13.63	5.46	7.50	4.38	3.27	3.41	**0.001 ***	**1.24 ^†^**	**0.001 ***	**2.27 ^†^**	**0.001 ***	**1.08 ^†^**
EDI-2: Ascetism	10.45	3.18	8.88	2.78	5.69	2.91	**0.001 ***	**0.53 ^†^**	**0.001 ***	**1.56 ^†^**	**0.001 ***	**1.12 ^†^**
EDI-2: Social Insecurity	13.40	4.54	9.60	4.10	4.51	3.03	**0.001 ***	**0.88 ^†^**	**0.001 ***	**2.30 ^†^**	**0.001 ***	**1.41 ^†^**
EDI-2: Total	152.13	27.03	127.14	20.64	81.45	22.16	**0.001 ***	**1.04 ^†^**	**0.001 ***	**2.86 ^†^**	**0.001 ***	**2.13 ^†^**
SCL-90-R: somatization	2.75	0.63	2.13	0.71	1.46	0.72	**0.001 ***	**0.93 ^†^**	**0.001 ***	**1.92 ^†^**	**0.001 ***	**0.94 ^†^**
SCL-90-R: obsess-comp.	2.83	0.59	2.34	0.57	1.53	0.64	**0.001 ***	**0.84 ^†^**	**0.001 ***	**2.11 ^†^**	**0.001 ***	**1.34 ^†^**
SCL-90-R: interpers.sens.	3.03	0.53	2.50	0.54	1.38	0.67	**0.001 ***	**1.00 ^†^**	**0.001 ***	**2.73 ^†^**	**0.001 ***	**1.84 ^†^**
SCL-90-R: depressive	3.24	0.43	2.68	0.53	1.67	0.62	**0.001 ***	**1.17 ^†^**	**0.001 ***	**2.93 ^†^**	**0.001 ***	**1.74 ^†^**
SCL-90-R: anxiety	2.76	0.55	1.96	0.64	1.08	0.54	**0.001 ***	**1.36 ^†^**	**0.001 ***	**3.09 ^†^**	**0.001 ***	**1.49 ^†^**
SCL-90-R: hostility	2.32	0.88	1.43	0.73	0.87	0.67	**0.001 ***	**1.10 ^†^**	**0.001 ***	**1.86 ^†^**	**0.001 ***	**0.81 ^†^**
SCL-90-R: phobic anxiety	2.04	0.83	1.32	0.79	0.51	0.52	**0.001 ***	**0.89 ^†^**	**0.001 ***	**2.22 ^†^**	**0.001 ***	**1.21 ^†^**
SCL-90-R: paranoia	2.48	0.62	1.57	0.59	0.87	0.53	**0.001 ***	**1.50 ^†^**	**0.001 ***	**2.77 ^†^**	**0.001 ***	**1.25 ^†^**
SCL-90-R: psychotic	2.31	0.52	1.56	0.45	0.85	0.48	**0.001 ***	**1.55 ^†^**	**0.001 ***	**2.92 ^†^**	**0.001 ***	**1.53 ^†^**
SCL-90-R: GSI index	2.73	0.36	2.06	0.33	1.26	0.38	**0.001 ***	**1.90 ^†^**	**0.001 ***	**3.96 ^†^**	**0.001 ***	**2.26 ^†^**
SCL-90-R: PST index	81.63	6.60	71.93	8.36	55.42	13.26	**0.001 ***	**1.29 ^†^**	**0.001 ***	**2.50 ^†^**	**0.001 ***	**1.49 ^†^**
SCL-90-R: PSDI index	3.01	0.34	2.60	0.36	2.03	0.36	**0.001 ***	**1.20 ^†^**	**0.001 ***	**2.81 ^†^**	**0.001 ***	**1.57 ^†^**
TCI-R: Novelty seeking	103.32	16.27	99.00	16.77	100.79	14.63	0.105	0.26	0.347	0.16	0.460	0.11
TCI-R: Harm avoidance	133.15	13.32	126.94	16.59	109.63	16.15	**0.018 ***	0.41	**0.001 ***	**1.59 ^†^**	**0.001 ***	**1.06 ^†^**
TCI-R: Reward-depend.	95.83	17.43	98.68	15.33	103.25	15.83	0.289	0.17	**0.007 ***	0.45	0.062	0.29
TCI-R: Persistence	103.42	22.25	102.77	20.58	107.33	19.98	0.852	0.03	0.267	0.19	0.149	0.23
TCI-R: Self-directedness	96.03	13.89	104.39	15.30	124.70	17.28	**0.002 ***	**0.57 ^†^**	**0.001 ***	**1.83 ^†^**	**0.001 ***	**1.24 ^†^**
TCI-R: Cooperativeness	124.40	19.00	133.38	16.53	139.31	11.39	**0.001 ***	**0.50 ^†^**	**0.001 ***	**0.95 ^†^**	**0.013 ***	0.42
TCI-R: Self-transcendence	73.35	13.87	63.43	15.43	63.73	17.46	**0.001 ***	**0.68 ^†^**	**0.001 ***	**0.61 ^†^**	0.903	0.02

Note. SD: standard deviation. * Bold: significant comparison (0.05 level). ^†^ Bold: effect size into the moderate (|*d*| > 0.50) to high range (|*d*| > 0.80). YFAS-2: Yale food addiction scale 2.0. BMI: Body mass index. EDI-2: Eating disorders Inventory 2. SCL-90-R: Symptom Checklist-90-Revised. TCI-R: Temperament and Character Inventory-Revised.

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
