# Peer review of "Food Addiction in Eating Disorders and Obesity: Analysis of Clusters and Implications for Treatment"

_nutrients, 2019, doi:10.3390/nu11112633_

Round 1

Reviewer 1 Report

Manuscript submitted for evaluation entitled „Food Addiction in Eating Disorders and Obesity: Analysis of Clusters and Implications for Treatment” is a valuable, well-written study which allowed the separation of clusters within the FA group using psychopathological symptomatology. Although research in this area has already been carried out, the above work adds new, significant knowledge. Below are some minor comments to the manuscript:

1.       Abbreviations should be explained first where they appear and if the abbreviation is used it should be entered in the first place of use (L120 is obesity and the abbreviation was given in L121 after the next use of the word) The full names of the questionnaires should be given in the first place where they appear with the appropriate abbreviation, if any (e.g. L143, 150… is full questionnaires name without abbreviation).

2.       In my opinion it would be more appropriate if both parts of Fig. 1 correspond with each other in the order of the factors indicated. In addition, adjusting the font size to fit on one line will allow data to be displayed in a single-line display for both panels. In its present form, Fig. 1 is difficult to interpretation. In addition, it should comply with the principle of self-explanation, so the abbreviations used should be developed in the description of Figure 1.

3.       Figure 2 needs improvement. There is no description of the Y axis and unnecessary multiple repetitions of "%" are used, which reduces the readability of the chart. Repetitions "%" from the Y axis should be moved to the description of the axis [%] and removed from the bars.

Author Response

Response to Reviewer 1 Comments

Thank you for the comments. We have made changes to the manuscript according to the reviewers’ comments. We have included a 'Revised Manuscript with Track Changes' where 'Track Changes' has been used to indicate where changes have been made (with the sole exception of tables). The manuscript has been prepared according to the journal's instructions. Our answers to the reviewer can be seen below.

Abbreviations should be explained first where they appear and if the abbreviation is used it should be entered in the first place of use (L120 is obesity and the abbreviation was given in L121 after the next use of the word) The full names of the questionnaires should be given in the first place where they appear with the appropriate abbreviation, if any (e.g. L143, 150… is full questionnaires name without abbreviation).

Response 1: As suggested by the reviewer, the abbreviations have been revised and corrected.

In my opinion it would be more appropriate if both parts of Fig. 1 correspond with each other in the order of the factors indicated. In addition, adjusting the font size to fit on one line will allow data to be displayed in a single-line display for both panels. In its present form, Fig. 1 is difficult to interpretation. In addition, it should comply with the principle of self-explanation, so the abbreviations used should be developed in the description of Figure 1

Response 2: We have revised the Figure 1 according to the reviewer’s suggestion.

Figure 2 needs improvement. There is no description of the Y axis and unnecessary multiple repetitions of "%" are used, which reduces the readability of the chart. Repetitions "%" from the Y axis should be moved to the description of the axis [%] and removed from the bars.

Response 3: We have revised the Figure 2 according to the reviewer’s suggestion.

Reviewer 2 Report

It is not clear if all the participants are obese or not.  Are there dietary intake data? It would be helpful to know the participants' habitual dietary intakes.  It would be helpful to include the male participants, although the sample size is relatively small, statistical analysis can still be done.  Are the demographical variables included in the analysis? 

Author Response

Response to Reviewer 2 Comments

Thank you for the comments. We have made changes to the manuscript according to the reviewers’ comments. We have included a 'Revised Manuscript with Track Changes' where 'Track Changes' has been used to indicate where changes have been made (with the sole exception of tables). The manuscript has been prepared according to the journal's instructions. Our answers to the reviewers can be seen below.

It is not clear if all the participants are obese or not.

Response 1: We understand the concern of the reviewer; therefore, the following statement has been added in the participants’ section of the manuscript:All participants into the OBE group reported BMI higher than 30. This condition was also met for n=42 participants into the BED group (84.0%), n=27 into the BN group (22.7%) and n=2 into the OSFED group (4.1%).”

Are there dietary intake data? It would be helpful to know the participants' habitual dietary intakes.

Response 2: We agree with the reviewer that it would be helpful to know the participants' habitual dietary intakes. However, this data is not available for this study. We are very thankful for this observation, and we will consider this suggestion for further studies.

It would be helpful to include the male participants, although the sample size is relatively small, statistical analysis can still be done.

Response 3: We understand the reviewer’s suggestion and we greatly appreciate this advice. However, the frequency of males in the sample was too small: only n=21 men met criteria to be included in the clustering and other statistical analysis. Like other multilevel analyses, the current cluster procedures based on likelihood estimation methods require that the number of clustering units be adequately large (mainly standard errors) to be estimated without bias (Maas, & Hox, 2005). In addition, although clustered data have been used for multiple samples sizes, the sample size of crucial variables is of utmost importance (Snijders, & Bosker, 2012). Simulation studies have shown that the standard procedures downwardly biased estimates when the number of clustering units is below 30-40 for binary outcomes (McNeish & Harring, 2017).

Maas, C.J., Hox,J.J. (2005). Sufficient sample sizes for multilevel modeling. Methodology: European Journal of Research Methods for the Behavioral and Social Sciences, 1, 86–92.

McNeish, D.M., & Harring, J.R. (2017). Clustered data with small sample sizes: comparing the performance of model-based and design-based approaches. Communications in Statistics, 46(2), 855-869.

Snijders, T.A.B., & Bosker, R.J. (2012). Multilevel analysis: an introduction to basic and advanced multilevel modeling. (2nd ed). London: Sage publishers.

Are the demographical variables included in the analysis?

Response 4: The clustering was carried out using the list of indicators detailed in the statistical analysis section, which measure the patients’ clinical profile: ED severity level (EDI-2 scales), global psychopathological state (SCL-90-R scales), personality profile (TCI-R scales) and diagnostic subtype. Sociodemographics were not considered as indicators. However, we considered interesting to obtain the comparison between the empirical clusters obtained in the frequency distribution of the sociodemographic measures for each cluster, and the potential differences between the groups (first block of Table 1).